# ELLA: Exploration through Learned Language Abstraction

**Suvir Mirchandani**
Computer Science
Stanford University
suvir@cs.stanford.edu

**Siddharth Karamcheti**
Computer Science
Stanford University
skaramcheti@cs.stanford.edu

**Dorsa Sadigh**
Computer Science and Electrical Engineering
Stanford University
dorsa@cs.stanford.edu

## Abstract

Building agents capable of understanding language instructions is critical to effective and robust human-AI collaboration. Recent work focuses on training these agents via reinforcement learning in environments with synthetic language; however, instructions often define long-horizon, sparse-reward tasks, and learning policies requires many episodes of experience. We introduce ELLA: Exploration through Learned Language Abstraction, a reward shaping approach geared towards boosting sample efficiency in sparse reward environments by correlating high-level instructions with simpler low-level constituents. ELLA has two key elements: 1) A termination classifier that identifies when agents complete low-level instructions, and 2) A relevance classifier that correlates low-level instructions with success on high-level tasks. We learn the termination classifier offline from pairs of instructions and terminal states. Notably, in departure from prior work in language and abstraction, we learn the relevance classifier online, *without* relying on an explicit decomposition of high-level instructions to low-level instructions. On a suite of complex BabyAI [11] environments with varying instruction complexities and reward sparsity, ELLA shows gains in sample efficiency relative to language-based shaping and traditional RL methods.

## 1 Introduction

A long-standing goal for robotics and embodied agents is to build systems that can perform tasks specified in natural language [1, 8, 20, 25, 38, 41, 42]. Central to the promise of language is its ability to cleanly specify complex, multi-step instructions. Instructions like *make a cup of coffee* define long-horizon tasks as *abstractions* over lower-level components—simple instructions like *pick up a cup* or *turn on the coffee maker*. Leveraging these abstractions can help amplify the sample efficiency and generalization potential of our autonomous agents.

One way to do this is through the lens of *instruction following*, which can be framed in several ways. One common framing—and the one we use in this work—is via reinforcement learning (RL): an agent is given a start state, a language instruction, and a corresponding reward function to optimize that usually denotes termination [20, 30]. While RL can be a useful framing, such approaches are often not sample efficient [11, 20]. Especially in the case of complex, highly compositional language instructions, RL agents can fail to make progress quickly—or at all. There are several reasons for poor performance in these settings; paramount is that in many environments, these instructions are

35th Conference on Neural Information Processing Systems (NeurIPS 2021).

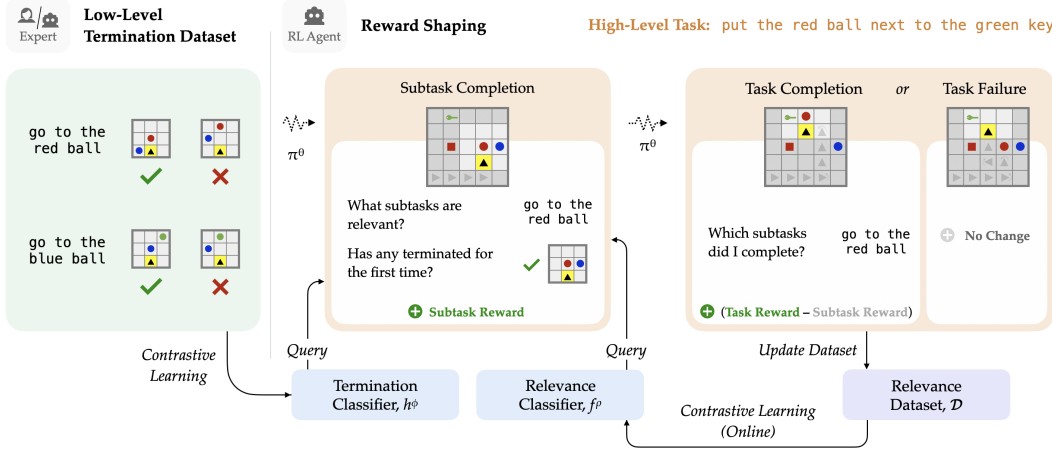

Figure 1: In this example, an expert provides examples of when low-level *go to* tasks are and are not solved, and the agent trains a termination classifier. During policy learning, the agent rewards itself for completing *go to* subtasks—specifically, those which are relevant to its current high-level task. It stores successful experiences to learn to correlate low-level *go to* tasks with high-level *put next* tasks.

tied to sparse reward functions that only provide signal upon completion of the high-level task, which drastically hurts sample efficiency. For example, seeing reward for *make a cup of coffee* would require already having turned on the coffee maker, which could itself be a difficult task. Such "bottleneck" states, which have zero intermediate reward, complicate exploration and learning [29, 39].

Our goal is to improve sample efficiency for instruction following agents operating in sparse reward settings. Driving our approach is using the principle of *abstraction*—the fact that complex instructions entail simpler ones—to guide exploration. Consider our coffee example; we would like to guide the agent to explore in a structured fashion, learning low-level behaviors first (*pick up a cup*, *turn on the coffee machine*) and building up to solving the high-level task. To do this, we frame our problem via *reward shaping*, the general technique of supplying auxiliary rewards to guide learning [31, 35].

Our approach, Exploration through Learned Language Abstraction (ELLA) provides intermediate rewards to an agent for completing relevant low-level behaviors as it tries to solve a complex, sparse reward task. Notably, our approach 1) learns to identify the low-level primitives helpful for a high-level language task *online*, and 2) does not require a strict hierarchical decomposition of language instructions to these primitives. Rather, ELLA uses low-level instructions to support agents performing complex tasks, bonusing agents as they complete *relevant* behaviors. In contrast to prior work [3, 23] that assumes strict contracts over instruction decomposition, our contract is simple and general. This prior work assumes each high-level task is comprised of exact series of low-level policies—an assumption that is only possible when the full set of primitives is known ahead of time, and fails when new actions or further exploration are necessary.

In order to bonus agents as they complete relevant subtasks, ELLA assumes access to a set of low-level instructions and corresponding termination states, similar to the data assumed in prior work [6]. While collecting such data for complex tasks may require time and effort, annotations for low-level instructions are more tractable. Humans can quickly annotate instructions like *pick up the cup* or *open the cupboard*, or even build tools for generating such examples. This is increasingly apparent in prior work [5, 22, 25, 43]; low-level instructions are used to simplify the problem of interpreting high-level instructions, through a variety of mechanisms. In our case, these low-level instructions provide the basis for reward shaping.

We empirically validate our abstraction-based reward shaping framework on a series of tasks via the BabyAI platform [11]. We compare against a standard RL baseline as well as to a strong language-based reward shaping approach [17], and find that our method leads to substantial gains in sample efficiency across a variety of instruction following tasks.

## 2 Related Work

We build on a large body of work that studies language in the context of reinforcement learning (RL) [27] with a focus on methods for instruction following, leveraging hierarchy, and reward shaping.

**Instruction Following.** Instruction following focuses on the automated execution of language commands in simulated or real environments [1, 16, 26, 41, 44]. We focus on the RL setting [11, 30], where agents learn language-conditioned policies in a Markov decision process (MDP).

Even simple instruction following problems are difficult to learn [28], which has made 2D environments important test-beds for new learning methods [3, 11, 46]. Such environments, which are procedurally-generated, are useful because they decouple exploration from the problem of perception, and demand policies that do not overfit to narrow regions in the state space [10, 13].

We use BabyAI [11] as the platform for our tasks. BabyAI's language is synthetic, similar to prior work examining RL for instruction following at scale [20, 21]. These environments are scalable starting points for new instruction following approaches, since deep RL remains sample-inefficient but has the potential to flexibly deal with complex environments [21]. Simplified environments also make investigating concepts such as language abstraction possible, and these concepts can generalize to real human-AI interactions [25, 43]. We describe our experimental setup in Section 5.

**Hierarchical and Language-Structured Policies.** A large body of work in hierarchical RL studies temporal abstractions of MDPs, where policies are designed or learned at multiple levels of hierarchy [40]. Recent works have made the connection to language: for example, high-level policies can act over the space of low-level instructions [23]. Other hierarchical language techniques include learning models that predict symbolic representations of reward functions at different granularities [5, 24], and compositional approaches like policy sketches [3] and modular control [15], methods that explicitly decompose high-level instructions into sequences of predefined low-level instructions, or subtasks.

In this work, we use language hierarchically, but *without* the strict decomposition assumption made by prior work: we do not assume high-level tasks are composed of an exact series of low-level tasks. Without this assumption, it is infeasible to use explicitly modular approaches [3, 5, 15]. A further distinguishing factor of our method relative to prior work is that we do not require high-level task decompositions be provided *a priori*. We expand on these features in Section 3.

**Language and Reward Shaping.** Reward shaping is a general technique for supplying auxiliary rewards in order to guide an agent's learning process. Certain types of reward transformations, such as potential-based rewards, do not change the optimal policy but can impact sample efficiency [31]. Prior work has applied reward shaping to language-based instruction following settings. Goyal et al. [17] train a classification network on (trajectory, language) data to predict if a trajectory (parameterized by action frequencies) matches a language description, and evaluate the network at every time step to form a potential-based reward shaping function. They evaluate their method on Montezuma's Revenge, which is a complex task, but with a static, deterministic layout that does not speak to the generalization potential of such methods. Relatedly, Waytowich et al. [45] use a narration-based approach, where high-level tasks are explicitly broken down into low-level tasks as narrations. Finally, Misra et al. [30] use a shaping function based on spatial distance to goals specified in language, but require the location of the goal to be known. In this work, we use the same high level principle—language can yield useful intermediate rewards—but do not make the restrictive assumptions of prior work about the environment or the way tasks are structured.

**Other Forms of Guidance.** Several other methods exist for improving the sample efficiency of RL in sparse reward settings. These methods reward an aspect of exploration that is orthogonal to our approach, and we see potential for combining these methods with ours. One popular technique is hindsight experience replay (HER), in which failed trajectories are stored in a replay buffer and relabeled with new goals that get nonzero reward [4]. Jiang et al. [23] extend the method to hierarchical language settings, assuming a mapping from states to corresponding goals. Cideron et al. [12] learn this mapping alongside policy training using the environment reward as supervision. Another technique for improving sample efficiency is through intrinsic motivation [9, 32, 34, 36]. In general, these methods instantiate reward shaping to incentivize accessing novel, diverse, or unpredictable parts of the state space via intrinsic rewards, and they can be extended to language-

conditioned settings as well [14]. In Appendix D, we evaluate the performance of RIDE [34], an intrinsic motivation method that rewards actions that cause significant changes to the state, and discuss potential synergies with ELLA.

## 3 Problem Statement

We consider an augmented MDP $\mathcal{M}$ defined by the tuple $(\mathcal{S}, \mathcal{A}, T, R, \mathcal{G}, \gamma)$ where $\mathcal{S}, \mathcal{A}, T$, and $\gamma$ are standard. $\mathcal{A}$ consists of primitive actions—in BabyAI, these are navigation primitives (`forward`, `pick up`, etc.). $\mathcal{G}$ is a set of language instructions, from which a high-level task instruction $g$ is drawn, and $R : \mathcal{S} \times \mathcal{A} \times \mathcal{G} \to [0, 1]$ represents the state-action reward given some $g$. Via RL, we wish to find some policy $\pi : \mathcal{S} \times \mathcal{G} \to \mathcal{A}$ that maximizes the expected discounted return.

We specifically consider cases where $\mathcal{M}$ has a finite horizon $H$ and $R$ is sparse with respect to goal-completion, making exploration difficult. Our aim is to construct a reward transformation $R \to R'$ which is policy invariant with respect to the original MDP (i.e. an optimal policy for $\mathcal{M}' = (\mathcal{S}, \mathcal{A}, T, R', g, \gamma)$ is also optimal in $\mathcal{M}$ and vice versa), while providing the agent strong signal conducive to sample-efficient exploration.

We assume access to a set of low-level instructions $\mathcal{G}_\ell$ such that every $g \in \mathcal{G}$ is *supported* by some $g_\ell \in \mathcal{G}_\ell$. Note that we do not require that high-level tasks fully factorize into low-level tasks. To clarify the distinction, say that $\mathcal{G}_\ell$ consists of *go to "x"* instructions: *go to the red ball*, *go to the blue ball*, etc. The instruction *go to the red ball and then go to the blue ball* can be fully factorized into low-level tasks in $\mathcal{G}_\ell$. On the other hand, the instruction *put the red ball next to the green key* is supported by *go to the red ball*, but it also requires picking up and putting down the ball—actions not covered by $\mathcal{G}_\ell$. Our framing permits exploration using a mixture of low-level instructions *as well as* primitive actions in $\mathcal{A}$ (such as `pick up` and `put down`).

We assume that examples of the corresponding termination states of instructions in $\mathcal{G}_\ell$ are easy to obtain—via a human expert or automation. This is reasonable for simple low-level tasks like the *go to* tasks in Figure 1. However, it is less feasible in domains where data collection is costly and the environment is hard to simulate; we address this further in Section 7.

The environments in our experiments are partially observable, so we use recurrent networks that ingest sequences of observations $(o_1, o_2, ..., o_t)$ rather than a single state $s_t$ [19]. Though our notation describes the fully observable setting, the extension to the partially observable case is straightforward.

## 4 ELLA

We present ELLA, our reward shaping framework for guiding exploration using learned language abstractions.[1] Figure 1 provides a graphical overview of ELLA. Our approach rewards an agent when it has completed low-level tasks that support a given high level task. To do this, it predicts 1) when a low-level task *terminates* and 2) when a low-level task is *relevant*. Section 4.1 describes our low-level termination classifier. Section 4.2 describes our relevance classifier, and how we learn it online during RL. Finally, Section 4.3 details our reward shaping scheme, which bonuses an agent for exploring states that satisfy relevant low-level tasks.

### 4.1 Learning the Low-Level Termination Classifier

We train a binary termination classifier $h^\phi : \mathcal{S} \times \mathcal{G}_\ell \to \{0, 1\}$ parameterized by $\phi$ to predict if a low-level task $g_\ell \in \mathcal{G}_\ell$ terminates in a particular state $s \in \mathcal{S}$. We assume access to $\mathcal{G}_\ell$ as well as positive and negative examples of low-level task termination states. This dataset could be annotated by a human or created automatically. While providing demonstrations of high-level tasks (e.g., *make a cup of coffee*) across varied environments is costly, it is more feasible to do so for low-level tasks (e.g., *pick up the mug*) which are shorter, simpler, and more generalizable. To represent $h^\phi$, we adapt the architecture from [11] which merges state and language representations using feature-wise linear modulation (FiLM) [33].

---

[1]Our code is available at `https://github.com/Stanford-ILIAD/ELLA`.

**Algorithm 1** Reward Shaping via ELLA

**Input:** Initial policy parameters $\theta_0$, relevance classifier parameters $\rho_0$, update rate $n$, low-level bonus $\lambda$, and RL optimizer OPTIMIZE
Initialize $\mathcal{D} \leftarrow \{(g : \mathcal{G}_\ell) \text{ for all } g \text{ in } \mathcal{G}\}$
**for** $k = 0, 1, \dots$ **do**
    Collect trajectories $\mathcal{D}_k$ using $\pi_k^\theta$.
    **for** $\tau \in \mathcal{D}_k$ **do**
        Set $N \leftarrow$ length of $\tau$
        Set $(r'_{1:N}, \hat{\mathbb{S}}) \leftarrow$ SHAPE$(\tau)$
        **if** $U(\tau) > 0$ **then**
            Set $r'_N \leftarrow$ NEUTRALIZE$(r'_{1:N})$
            Set $\mathcal{D}[g] \leftarrow$ UPDATEDECOMP$(\mathcal{D}, \hat{\mathbb{S}})$
    Update $\theta_{k+1} \leftarrow$ OPTIMIZE$(r'_{1:N})$.
    **if** $k$ is a multiple of $n$ **then**
        Update $\rho_{k+1}$ by optimizing cross entropy loss on a balanced sample from $\mathcal{D}$.

**function** SHAPE$(\tau)$
    Set $\hat{\mathbb{S}} \leftarrow \emptyset$
    **for** $g_\ell \in \mathcal{G}_\ell$ **do**
        **for** $g, (s_t, r_t) \in \tau$ **do**
            **if** $h^\phi(s_t, g_\ell) = 1$ and $g_\ell \notin \hat{\mathbb{S}}$ **then**
                Update $\hat{\mathbb{S}} \leftarrow \hat{\mathbb{S}} \cup \{g_\ell\}$
                Set $r'_t \leftarrow r_t + \lambda \cdot \mathbb{I}[f^\rho(g, g_\ell) = 1]$
    **return** $(r'_{1:N}, \hat{\mathbb{S}})$

**function** NEUTRALIZE$(r'_{1:N})$
    Set $T_{\mathbb{S}} \leftarrow \{t \mid 1 \le t \le N, r'_t > 0\}$
    **return** $r'_N - \sum_{t \in T_{\mathbb{S}}} \gamma^{t-N} \lambda$

**function** UPDATEDECOMP$(\mathcal{D}, \hat{\mathbb{S}})$
    Set $\mathbb{S} \leftarrow \mathcal{D}[g]$
    **return** $\mathbb{S} \cap \hat{\mathbb{S}}$ (or $\hat{\mathbb{S}}$ if $\mathbb{S} \cap \hat{\mathbb{S}} = \emptyset$)

## 4.2 Learning the Relevance Classifier

Our reward shaping approach also needs to evaluate relevance of a low-level instruction in addition to being able to classify termination. The relevance mapping from $\mathcal{G}$ to $\mathcal{P}(\mathcal{G}_\ell)$ (the power set of $\mathcal{G}_\ell$) is initially unknown to the agent. We refer to the output of this mapping as a decomposition of $g$, but note again that it need not be a *strict* decomposition. We propose that this mapping is represented by a separate binary relevance classifier $f^\rho : \mathcal{G} \times \mathcal{G}_\ell \to \{0, 1\}$ learned during policy training from a dataset of decompositions $\mathcal{D}$ collected online. The output of $f^\rho$ indicates whether a particular $g_\ell \in \mathcal{G}_\ell$ is relevant to some $g \in \mathcal{G}$. We use a Siamese network to represent $f^\rho$.

**Training the Relevance Classifier.** Suppose $\mathcal{D}$ already contains key-value pairs $(g, \mathbb{S})$ where $g \in \mathcal{G}$ and $\mathbb{S} \in \mathcal{P}(\mathcal{G}_\ell)$. Each $\mathbb{S}$ is an estimate of the oracle decomposition $\bar{\mathbb{S}}$ of a particular $g$. For every $g$ in $\mathcal{D}$, we enumerate negative examples of relevant subtasks from $\mathcal{G}_\ell \setminus \mathbb{S}$ and accordingly oversample positive examples from $\mathbb{S}$; we then optimize a cross entropy objective on this balanced dataset.

**Collecting Relevance Data Online.** We describe any trajectory that solves the high-level task as *successful*, regardless of whether they do so optimally, and any other trajectories as *unsuccessful*. For every successful trajectory of some $g$ we encounter during RL training, we record the low-level instructions which terminated. That is, we relabel the steps in the trajectory and use $h^\phi$ to determine which low-level instructions were completed (if any), yielding an estimate $\hat{\mathbb{S}}$ of the true decomposition $\bar{\mathbb{S}}$. If $\mathcal{D}$ already contains an estimate $\mathbb{S}$ for $g$, we simply deduplicate the multiple decomposition estimates by intersecting $\mathbb{S}$ and $\hat{\mathbb{S}}$. As the agent completes more successful trajectories, decompositions in $\mathcal{D}$ are more likely to get revisited.

**Intuition for Deduplication.** Intuitively, this method for deduplication denoises run-specific experiences, and distills the estimate $\mathbb{S}$ in $\mathcal{D}$ for a given $g$ to only those $g_\ell$ that are shared among multiple successful runs of $g$. Consider $g =$ *put the red ball next to the blue box*; successful runs early in training could highlight *go to the red ball* along with irrelevant *go to* low-level instructions. As $\pi^\theta$ improves, so do the decomposition estimates. Later runs may highlight only *go to the red ball*. Taking the intersection would yield only *go to the red ball* as our final estimate $\mathbb{S}$. This approach also helps to deal with false positives of $h^\phi$ by effectively boosting the estimate across runs. If deduplication produces a null intersection, which may happen due to false negatives of $h^\phi$, we err on the side of using the latest estimate. If we assume $h^\phi$ has perfect accuracy, the intersection will never be null, and the continual intersections will reflect the set of completed $g_\ell$ common to all successful runs of $g$. For an initialization conducive to deduplication and that provides $\pi^\theta$ strong incentive to complete low-level tasks from the beginning of training, we initialize $f^\rho$ to predict that $\mathbb{S} = \mathcal{G}_\ell$ for all $g$.

| Low-Level Task | Example |
|---|---|
| GOTO-ROOM (G1) | *go to a purple ball* |
| GOTO-MAZE (G2) | *go to a purple ball* |
| OPEN-MAZE (O2) | *open the red door* |
| PICK-MAZE (P2) | *pick up the blue box* |

| High-Level Task | Example(s) | $\mathcal{G}_\ell$ |
|---|---|---|
| PUTNEXT-ROOM | *put the purple ball next to the blue key* | G1 |
| PUTNEXT-MAZE | *put the purple ball next to the blue key* | G2 |
| UNLOCK-MAZE | *open the green door* | P2 |
| OPEN&PICK-MAZE | *open the red door and pick up the blue box* | O2, G2 |
| COMBO-MAZE | *put next, open, pick* | G2 |
| SEQUENCE-MAZE | *pick up a purple key and then put the grey box next to the red box* | G2 |

Table 1: Sample low- and high-level instructions.

### 4.3 Shaping Rewards

We now describe the shaped reward function $R'$ which relies on $h^\phi$ and $f^\rho$. $R'$ provides a bonus $\lambda$ whenever a relevant low-level task terminates. We do this by relabeling the current $(s_t, a_t, r_t, s_{t+1})$ with every $g_\ell$ and using $h^\phi$ to predict low-level task termination, and $f^\rho$ to predict relevance.

Critically, we do not want the shaping transformation from $R \to R'$ to be vulnerable to the agent getting "distracted" by the shaped reward (reward hacking), as in Randløv and Alstrøm [35] where an agent learns to ride a bicycle in a circle around a goal repeatedly accruing reward. To this end, we cap the bonus per low-level instruction per episode to $\lambda$ to prevent cycles. More generally, we want $R'$ to be *policy invariant* with respect to $\mathcal{M}$: it should not introduce any new optimal policies, and it should retain all the optimal policies from $\mathcal{M}$. Policy invariance is a useful property [31]: we do not want to reward a suboptimal trajectory (such as one that repeatedly completes a low-level instruction) more than an optimal trajectory in $\mathcal{M}$, and policy invariance guarantees this.

We establish two desiderata for $R'$ [7]: (a) The reward transformation should be policy invariant with respect to $\mathcal{M}$ for states from which the task is solvable, and (b) $R'$ should improve sample efficiency by encouraging subtask-based exploration. To satisfy our desiderata, we choose $R'$ such that successful trajectories get the same return under $\mathcal{M}$ and $\mathcal{M}'$, and that unsuccessful trajectories get lower returns under $\mathcal{M}'$ than trajectories optimal in $\mathcal{M}$. We will describe and intuitively justify each of these choices with respect to (a) and (b).

**Neutralization in Successful Trajectories.** We neutralize shaped reward in successful trajectories— that is, subtract the shaped reward at the final time step—so that successful trajectories gets the same return under both $R'$ and $R$.

More specifically, let $U(\tau) := \sum_{t=1}^N \gamma^t R(s_t, a_t)$, the cumulative discounted return under $R$ of a successful trajectory $\tau$. Let $U'(\tau)$ be likewise defined for $R'$. If $\tau$ has $N$ steps, the sparse reward under $R$ is $U(\tau) = \gamma^N r_N$. Under $R'$, if $T_\mathbb{S}$ is the set of the time steps at which a $\lambda$ bonus is applied, we set $r'_N$ to $r_N - \sum_{t \in T_\mathbb{S}} \gamma^{t-N} \lambda$. This is the value required to neutralize the intermediate rewards, such that $U'(\tau) = \gamma^N r_N = U(\tau)$. (Note that we cap the bonus per time step to $\lambda$—if multiple low-level language instructions terminate at a single state, only a bonus of $\lambda$ is applied.)

Theoretically, we could apply neutralization to *all* trajectories, not just successful ones, and we would satisfy property (a) [18]. However, this is harmful to property (b), because unsuccessful trajectories would result in zero return: a negative reward at the last time step would negate the subtask rewards, potentially hurting boosts in sample efficiency.

**Tuning $\lambda$ to Limit Return in Unsuccessful Trajectories.** Any successful trajectory gets the same return under $R'$ as under $R$ because of neutralization. By choosing $\lambda$ carefully, we can additionally satisfy the property that any unsuccessful trajectory gets a lower return under $R'$ than any trajectory selected by an optimal policy $\pi^*_{\mathcal{M}}$ in $\mathcal{M}$.

To achieve this, we need $\lambda$ to be sufficiently small. Assume that a trajectory $\tau$ selected by $\pi^*_{\mathcal{M}}$ takes no longer than $M$ time steps to solve any $g$. In the worst case, $M = H$, and $U(\tau)$ would be $\gamma^H r_H$. If $T_{\mathbb{S}}$ is the set of the time steps at which $R'$ provides $\lambda$ bonuses, $U'(\tau)$ would be $\lambda \sum_{t \in T_{\mathbb{S}}} \gamma^t$. We can upper bound $\sum_{t \in T_{\mathbb{S}}} \gamma^t$ with $|\mathcal{G}_\ell|$. Then, the following inequality is sufficient for maintaining (a):

$$\lambda < \frac{\gamma^H r_H}{|\mathcal{G}_\ell|}, \tag{1}$$

where $r_H$ is the value of a sparse reward if it were attained at time step $H$. We provide a more thorough justification of this choice of $\lambda$ in Appendix F. Note that $\lambda$ and the sparse reward can be both scaled by a constant if $\lambda$ is otherwise too small to propagate as a reward.

An important feature of our work is the realization that we can make this bound less conservative through minimal knowledge about the optimal policy and the environment (e.g., knowledge about the expected task horizon $M$, or a tighter estimate of the number of subtasks available in an environment instance). Such reasoning yields a feasible range of values for $\lambda$, and these can empirically lead to faster learning (Section 6.1). At a minimum, reasoning over the value of $\lambda$ using via (1) provides a generalizable way to incorporate this technique across a variety of different settings in future work.

We summarize our shaping procedure in Algorithm 1, with an expanded version in Appendix C.

## 5   Experiments

**Experimental Setup.** We run our experiments in BabyAI [11], a grid world platform for instruction following, where an agent has a limited range of view and receives goal instructions such as *go to the red ball* or *open a blue door*. Grid worlds can consist of multiple rooms connected by a closed/locked door (e.g., the UNLOCK-MAZE environment in Figure 2). The action space $\mathcal{A}$ consists of several navigation primitives (`forward`, `pickup`, etc.). Every task instance includes randomly placed distractor objects that agents can interact with. Rewards in BabyAI are sparse: agents receive a reward of $1 - 0.9\frac{t}{H}$ where $t$ is the time step upon succeeding at the high-level goal. If the goal is not reached, the reward is 0. By default, all rewards are scaled up by a constant factor of 20.

We evaluate our reward shaping framework using Proximal Policy Optimization (PPO) [37], but note that ELLA is agnostic to the RL algorithm used. We compare to PPO without shaping, as well as to LEARN, a prior method on language-based reward shaping [17] that provides rewards based on the predicted relevance of action frequencies in the current trajectory to the current instruction.

We focus on several high- and low-level tasks. Table 1 describes each task with examples. ROOM levels consist of a single $7 \times 7$ grid, while MAZE levels contain two such rooms, connected by a door; agents may need to open and pass through the door multiple times to complete the task. In the UNLOCK environment, the door is "locked," requiring the agent to hold a key of the same color as the door before opening it, introducing significant bottleneck states [29, 39]. We choose six high- and low-level task pairs in order to differ along three axes: *sparsity* of the high-level task, *similarity* of the low- and high-level tasks, and *compositionality* of the tasks in $\mathcal{G}$—the number of $g_\ell \in \mathcal{G}_\ell$ relevant to some $g$. We use these axes to frame our understanding of how ELLA performs in different situations.

We used a combination of NVIDIA Titan and Tesla T4 GPUs to train our models. We ran 3 seeds for each of the 3 methods in each environment, with runs taking 1 to 6 days.

**Results.** Figure 2 presents learning curves for ELLA, LEARN, and PPO (without shaping) across the six environments. We explain our results in the context of the three axes described above.

$\rightarrow$ *How does ELLA perform on tasks with different degrees of sparsity?*

In both single room (PUTNEXT-ROOM) and two room (PUTNEXT-MAZE) environments, ELLA induces gains in sample efficiency, using GOTO as $\mathcal{G}_\ell$. Relative gains are larger for the bigger

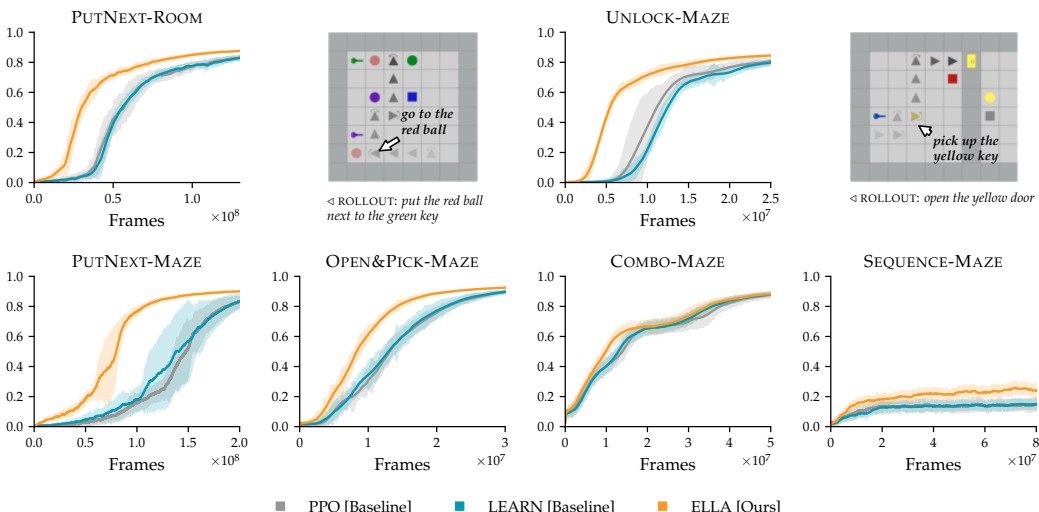

Figure 2: Average reward for ELLA and baselines in six environments, with error regions to indicate standard deviation over three random seeds. Example rollouts in the PUTNEXT-ROOM environment and UNLOCK-MAZE environment illustrate the final policy learned via ELLA, where the agent completes relevant low-level subtasks in order to solve the high-level task.

environment because reward signals are more spatially sparse and so spatial subgoals—visiting relevant objects—help the agent navigate.

Another degree of sparsity comes from bottleneck states: for example, in the UNLOCK environment, the agent must pick up a key of the corresponding color (the bottleneck) before it can successfully open a colored door and see reward. Without shaping, random exploration rarely passes such bottlenecks. However, in this experiment, ELLA rewards picking up keys, via the PICK low-level instruction, quickly learning to correlate picking up keys of the correct color with successfully unlocking doors, allowing agents to navigate these bottlenecks and improving sample efficiency.

→ *How does ELLA perform when the low-level tasks are similar to the high-level task?*

Tasks in the COMBO environment consist of multiple instruction types (e.g., *put the red ball next to the green box*, *open the green door*, and *pick up the blue box*). The latter instructions require minimal exploration beyond "going to" an object — such as executing an additional `open` or `pick up` primitive actions. That is, the low-level task GOTO is more similar to this high-level task set than in the other environments such as PUTNEXT. As a result, ELLA does not increase in sample efficiency in COMBO. However, it notably does not perform *worse* than baselines: the exploration it encourages is not harmful, but is simply not helpful. This is expected as achieving the COMBO tasks is about as difficult as exploring the relevant GOTO subtasks.

→ *How does ELLA perform when the high-level tasks are compositional and diverse?*

We test ELLA on OPEN&PICK using *two* low-level instruction families, OPEN and GOTO. The instruction *open the yellow door and pick up the blue box* abstracts over *go to the yellow door*, *open the yellow door*, and *go to the blue box*. Although learning $f^\rho$ becomes harder with more compositional instructions, the boost in sample efficiency provided by ELLA remains.

Similarly, SEQUENCE has a combinatorially-large number of compositional instructions: it requires two of *put next*, *open*, and *pick up* in the correct order. It has over 1.5 million instructions compared to the other high-level tasks with 300 to 1500 instructions. Although the exploration problem is very difficult, we see marginal gains from ELLA; we discuss this further in Section 7.

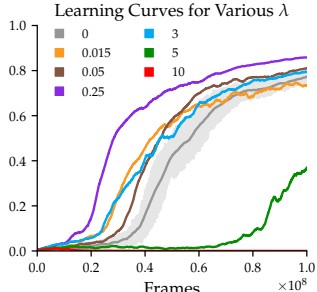
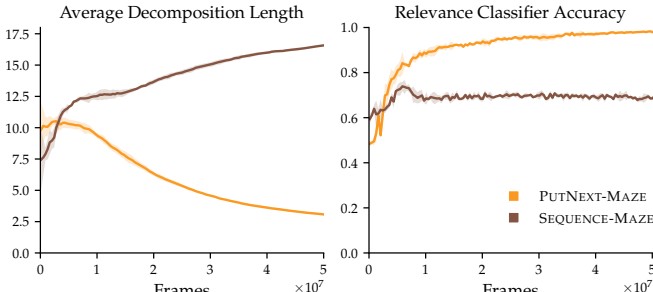

Figure 3: Average reward in PUTNEXT-ROOM for various values of $\lambda$. Smaller values are more conservative but potentially less helpful than larger values. $\lambda = 0$ indicates no shaping; for this case, we show an error region over three random seeds.

Figure 4: Two metrics to analyze ELLA's performance over an RL run are the average number of low-level instructions per high-level instruction in $\mathcal{D}$ and the accuracy of the relevance classifier $f^\rho$ on a balanced validation set. For PUTNEXT-MAZE, $f^\rho$ improves as decomposition estimates get less noisy. For the challenging SEQUENCE-MAZE task, ELLA still incentivizes subtask completion, but the decomposition estimates are noisy and $f^\rho$ is less accurate.

## 6 Analyzing ELLA

In this section, we analyze two aspects of ELLA: the effect of different choices for the low-level reward hyperparameter (Section 6.1), and the performance of the relevance classifier (Section 6.2).

### 6.1 Effect of the Low-Level Task Reward

The magnitude of the low-level reward $\lambda$ is a critical hyperparameter. Instead of using ad hoc methods for tuning $\lambda$, Section 4.3 provides guidelines for choosing a reasonable range for $\lambda$. To summarize, we want to pick $\lambda$ small enough that any unsuccessful trajectory does not receive greater return than any optimal trajectory. In this section, we examine PUTNEXT-ROOM and evaluate various $\lambda$ values determined using the multi-tier assumptions alluded to in Section 4.3. For this task, we have that $H = 128$ and $|\mathcal{G}_\ell| = 36$. With no knowledge of the environment or optimal policy, the loosest bounds for $M$ and $|T_\mathbb{S}|$ ($H$ and $|\mathcal{G}_\ell|$ respectively), as in (1) yield $\lambda = 0.015$. With the minimal assumption that an optimal policy solves the task in under 100 steps, we arrive at $\lambda = 0.05$. For 40 steps, we arrive at $\lambda = 0.25$. Figure 3 compares learning curves for $\lambda$—the smallest being more conservative and not having a great effect on sample efficiency, the middle values illustrating the value of rewarding relevant subtasks, and the largest values chosen specifically to demonstrate that egregious violations of policy invariance can "distract" the agent from the high-level task and lead to unstable learning. The results in Section 5 use $\lambda = 0.25$ for PUTNEXT, UNLOCK, and COMBO, and use $\lambda = 0.5$ for OPEN&PICK and SEQUENCE (which have longer horizons $H$).

### 6.2 Progress of the Relevance Classifier

The online relevance dataset and classifier provide transparency into the type of guidance that ELLA provides. Figure 4 focuses on PUTNEXT-MAZE and SEQUENCE-MAZE, and shows two metrics: the average number of subtasks per high-level instruction in $\mathcal{D}$, which we expect to decrease as decomposition estimates improve; and the validation accuracy of $f^\rho$ in classifying a balanced oracle set of subtask decompositions, which we expect to increase if ELLA learns language abstractions.

For PUTNEXT-MAZE, the number of subtasks per high-level instruction decreases and the relevance classifier becomes very accurate. This happens as, from Figure 2, the policy improves in terms of average return. For the SEQUENCE-MAZE task, ELLA has marginal gains in performance compared to baselines (Figure 2). This is likely due to ELLA incentivizing general subtask completion, evidenced by the growing average length of the decompositions. However, the large number of diverse language instructions in SEQUENCE-MAZE makes it difficult to estimate good decompositions via our deduplication method (as it is uncommon to see the same instruction multiple times). It is difficult to learn $f^\rho$ online and offer the most targeted low-level bonuses in this case.

# 7 Discussion

We introduce ELLA, a reward shaping approach for guiding exploration based on the principle of abstraction in language. Our approach incentivizes low-level behaviors without assuming a strict hierarchy between high- and low-level language, and learns to identify relevant abstractions online.

As noted in Section 2, several other methods exist for addressing sample inefficiency in sparse reward settings. While we focus on using language as the basis for reward shaping, intrinsic rewards can also be based on the dynamics of the environment. For example, RIDE [34] provides rewards for actions that produce significant changes to the state. In Appendix D, we discuss how ELLA can synergize with such methods to reward different aspects of exploration.

**Limitations and Future Work.** Our framework requires termination states for each low-level instruction. This assumption is similar to that made by Jiang et al. [23] and Waytowich et al. [45], and is weaker than the assumption made by the LEARN baseline of Goyal et al. [17], where full demonstrations are required. Even so, training accurate termination classifiers can require many examples—in this work, we used 15K positive and negative pairs for each low-level task. In Appendix E, we provide an ablation on the data budget for the termination classifier. Our approach is less feasible in domains where collecting examples of low-level terminating conditions is costly.

Highly compositional and diverse instructions, as in the SEQUENCE-MAZE environment, remain a challenge. Data augmentation (e.g., GECA [2]) on the online relevance dataset could potentially improve performance on these large instruction sets. Additional boosts in performance and generalization could emerge from instilling language priors via offline pre-training. More sophisticated relevance classifiers may also prove helpful: instead of deduplicating and training on $\mathcal{D}$, we could maintain a belief distribution over whether high- and low-level instructions are relevant conditioned on trajectories observed online. Furthermore, the relevance classifier could be extended to model the temporal order of the low-level tasks.

Our approach shows strong performance in several sparse reward task settings with synthetic language. Extensions to natural language could model a broad spectrum of abstraction beyond two tiers.

## Acknowledgments and Disclosure of Funding

This project was supported by NSF Award 1941722 and 2006388, the Future of Life Institute, the Office of Naval Research, and the Air Force Office of Scientific Research. Siddharth Karamcheti is additionally supported by the Open Philanthropy AI Fellowship.

Furthermore, we would like to thank Dilip Arumugam, Suneel Belkhale, Erdem Bıyık, Ian Huang, Minae Kwon, Ben Newman, Andrea Zanette, and our anonymous reviewers for feedback on earlier versions of this paper.

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
