# A  Experiment Details

Table 1 provides additional reference information about our suite of evaluation tasks.

Table 1: Details of our low- and high-level tasks.

| Low-Level Task | # Instr. | Example | |
|---|---|---|---|
| GoTo-Room (G1) | 36 | *go to a yellow ball* |  |
| GoTo-Maze (G2) | 42 | *go to a red key* |  |
| Open-Maze (O2) | 6 | *open the green door* |  |
| Pick-Maze (P2) | 36 | *pick up a red box* |  |

| High-Level Task | # Instr. | Example | Visualization | $\mathcal{G}_\ell$ |
|---|---|---|---|---|
| PutNext-Room | 306 | *put the blue key next to the yellow ball* |  | G1 |
| PutNext-Maze | 1440 | *put the yellow ball next to a purple key* |  | G2 |
| Unlock-Maze | 6 | *open the green door* |  | P2 |
| Open&Pick-Maze | 216 | *open the yellow door and pick up the grey ball* |  | O2, G2 |
| Combo-Maze | 1266 | *pick up the green ball* |  | G2 |
| Sequence-Maze | >1M | *open the grey door after you put the yellow ball next to a purple key* |  | G2 |

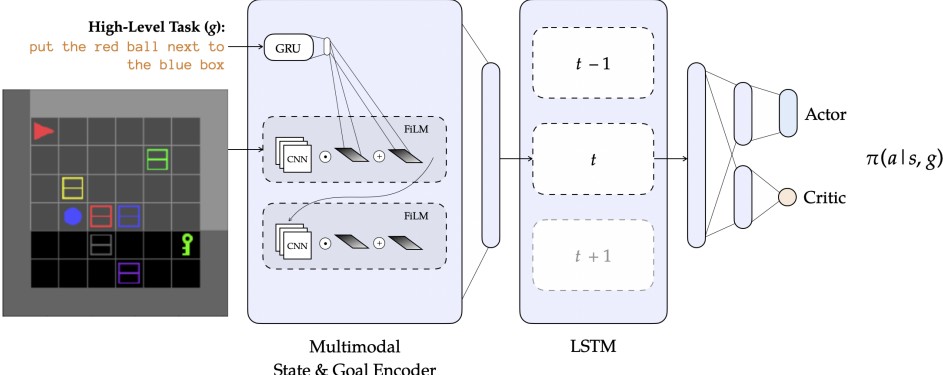

Figure 1: The actor-critic architecture processes an observation and language instruction using a multimodal encoder based on feature-wise linear modulation (FiLM) [6]. It then uses an LSTM to recurrently process a history of observations and projects the output onto actor and critic heads.

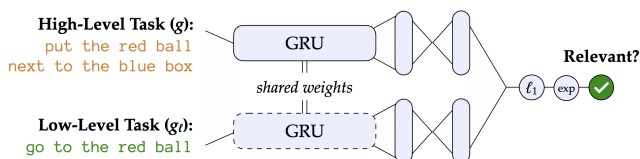

Figure 2: The relevance classifier consists of a Siamese network that returns a binary prediction of whether a low-level instruction is relevant to a high-level instruction.

## B   Training Details

We adapt the PPO implementation from [2] with the default hyperparameters (discount factor of $0.99$, learning rate (via Adam) of $7 \times 10^{-4}$, batch size of $2560$, minibatch size of $1280$, entropy coefficient of $0.01$, value loss coefficient of $0.5$, clipping-$\epsilon$ of $0.2$, and generalized advantage estimation parameter of $0.99$). We use the actor-critic architecture from [2] (Figure 1).

### B.1   Termination and Relevance Classifiers

The termination classifier $h^\phi$ is an adapted version of the architecture in Figure 1 that uses a binary prediction head instead of the actor and critic heads. Our implementation of $h^\phi$ makes predictions based on single observations; in our BabyAI tasks, final observations are sufficient for evaluating whether a task has terminated.

To train $h^\phi$, we require positive and negative examples of states at which low-level language instructions terminate. We use the automated expert built into BabyAI to generate 15K low-level episodes. For each episode, we use the final observation as a positive example for that episode's language instruction and a randomly sampled state from the trajectory as a negative example. Similarly, we generate 200 episodes for validation data. We augment the datasets with additional negative examples by sampling 35 mismatching low-level instructions for each terminating observation. We use a batch size of $2560$ and optimize via Adam with a learning rate of $10^{-4}$. We train for 5 epochs and use the iteration that achieves the highest validation accuracy.

We train the relevance classifier $f^\rho$ online with the architecture described in Figure 2. For an initialization conducive to deduplication, we initialize $f^\rho$ to predict that any $g_\ell$ is relevant to any $g$. To do this, we randomly sample 100 high-level instructions, cross that set with $\mathcal{G}_\ell$, label the pairs as relevant, and train for 20 epochs. We use a learning rate of $10^{-4}$ and a batch size of $10$. Online updates based on the online dataset $\mathcal{D}$ involve 3 gradient updates to $\rho$ for every 50 iterations of PPO.

## B.2   LEARN

We will now detail LEARN [3] which we use as a baseline. LEARN rewards trajectories that it predicts are relevant to a given language instruction. We reimplement LEARN based on an open-source implementation.[1] We collect 15K episodes of BabyAI's automated expert completing low-level tasks and process them to create (*instruction*, *action frequency*) data. We use this dataset to train the classifier used in LEARN which predicts whether the action frequencies of the current trajectory are related to the current instruction. The coefficient on the shaped rewards is a hyperparameter; based on an informed sweep of values, we set its value to $0.01$. We train the classifier for 100 epochs.

An intuition for why LEARN has strong performance in static environments as in [4] but not in our setting is that the method requires action frequencies to provide a signal about whether the current trajectory is related to the language instruction. Our environments are dynamic, and so individual actions are less correlated to language tasks. Additionally, in our setting, the instructions during RL are high-level instructions which are selected from a different distribution than the low-level instructions available for training the classifier.

## B.3   RIDE

As an additional comparison point to ELLA, we reimplement the RIDE method [7] based on an open-source implementation.[2] RIDE rewards "impactful" changes to the state; we discuss the method further in Appendix D. We use the same architectures as the original work for the forward dynamics model, inverse dynamics model, and state embedding model. For comparability with our implementation of ELLA with PPO, we adapt RIDE to the on-policy setting by updating the dynamics models once per batch of on-policy rollouts. For hyperparameters, we use the values published in the code repository for the coefficients on forward dynamics loss and inverse dynamics loss (10 and $0.1$ respectively), as well as the published value for learning rate of $10^{-4}$. We tune the intrinsic reward coefficient (which we call $\lambda_R$) within the set $\{0.1, 0.5, 1\}$.

# C   Algorithm (Expanded)

Algorithm 1 breaks down ELLA in detail.

---

[1]https://github.com/prasoongoyal/rl-learn
[2]https://github.com/facebookresearch/impact-driven-exploration

---

**Algorithm 1** Reward Shaping with ELLA

---

1: **Input:** Initial policy parameters $\theta_0$, relevance classIfier parameters $\rho_0$, update rate $n$, low-level bonus $\lambda$, and on-policy RL optimizer OPTIMIZE
2: Initialize $\mathcal{D} \leftarrow \{(g : \mathcal{G}_\ell) \text{ for all } g \text{ in } \mathcal{G}\}$
3: **for** $k = 0, 1, \dots$ **do**
4:     Collect trajectories $\mathcal{D}_k = \{\tau_i\}$ using $\pi_k^\theta$.
5:     **for** $\tau \in \mathcal{D}_k$ **do**
6:         Set $N \leftarrow$ length of $\tau$
7:         Set $(r'_{1:N}, \hat{\mathbb{S}}) \leftarrow \text{SHAPE}(\tau)$
8:         **if** $U(\tau) > 0$ **then**                               ▷ If trajectory was successful
9:             Set $r'_N \leftarrow \text{NEUTRALIZE}(r'_{1:N})$
10:             Set $\mathcal{D}[g] \leftarrow \text{UPDATEDECOMP}(\mathcal{D}, \hat{\mathbb{S}})$
11:     Update $\theta_{k+1} \leftarrow \text{OPTIMIZE}(r'_{1:N})$.
12:     **if** $k$ is a multiple of $n$ **then**
13:         $\mathcal{D}' \leftarrow$ Sample positive and negative examples of relevant pairs $(g, g_\ell)$ from $\mathcal{D}$
14:         Update $\rho_{k+1} \leftarrow$ Optimize cross entropy loss on $\mathcal{D}'$
15:
16: **function** SHAPE($\tau$)
17:     Set $\hat{\mathbb{S}} \leftarrow \emptyset$
18:     **for** $g_\ell \in \mathcal{G}_\ell$ **do**
19:         **for** $g, (s_t, r_t) \in \tau$ **do**
20:             **if** $h^\phi(s_t, g_\ell) = 1$ and $g_\ell \notin \hat{\mathbb{S}}$ **then**      ▷ If $g_\ell$ has terminated for the first time
21:                 Update $\hat{\mathbb{S}} \leftarrow \hat{\mathbb{S}} \cup \{g_\ell\}$              ▷ Record $g_\ell$ in the decomposition
22:                 **if** $f^\rho(g, g_\ell) = 1$ **then**                             ▷ If $g_\ell$ is relevant
23:                     Set $r'_t = r_t + \lambda$                              ▷ Apply low-level bonus
24:     **return** $(r'_{1:N}, \hat{\mathbb{S}})$
25:
26: **function** NEUTRALIZE($r'_{1:N}$)
27:     Set $T_\mathbb{S} \leftarrow \{t \mid 1 \le t \le N, r'_t > 0\}$          ▷ Get time steps at which rewards were shaped
28:     **return** $r'_N - \sum_{t \in T_\mathbb{S}} \gamma^{t-N} \lambda$      ▷ Final reward neutralizes shaped rewards (Section 4.3)
29:
30: **function** UPDATEDECOMP($\mathcal{D}$ $\hat{\mathbb{S}}$))
31:     Set $\mathbb{S} \leftarrow \mathcal{D}[g]$
32:     **if** $\mathbb{S} \cap \hat{\mathbb{S}} = \emptyset$ **then**
33:         **return** $\hat{\mathbb{S}}$
34:     **else**
35:         **return** $\mathbb{S} \cap \hat{\mathbb{S}}$

---

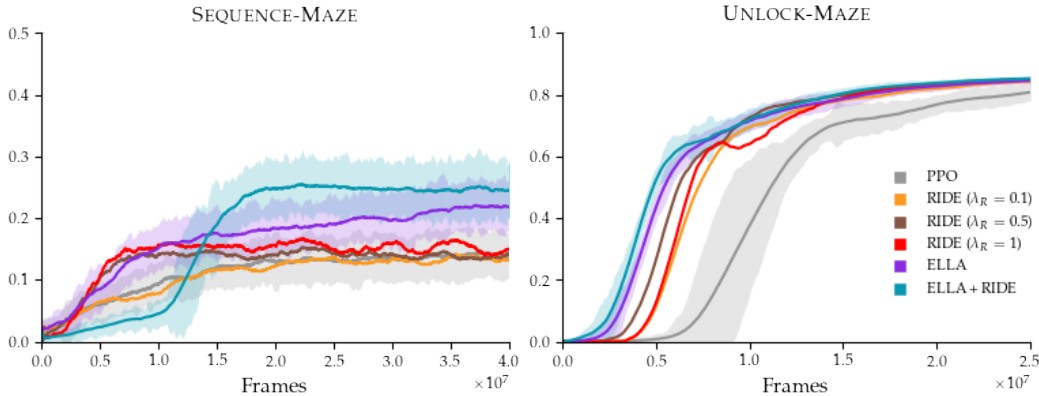

Figure 3: Learning curves for PPO (no shaping), ELLA, RIDE, and ELLA+RIDE in the SEQUENCE-MAZE task. For PPO, ELLA, and ELLA+RIDE, translucent regions show standard deviation of return over three random seeds.

## D   Relation to Intrinsic Motivation

As mentioned in Section 2, curiosity and intrinsic motivation methods use reward shaping to incentivize exploration to novel, diverse, or unpredictable parts of the state space [1, 5, 7, 8]. We adapt RIDE [7], an intrinsic motivation method, to our setting and discuss one way in which ELLA could be combined with intrinsic motivation methods.

RIDE rewards actions that produce "impactful" changes to its representation of the state. The state representation function is learned via a forward dynamics model and inverse dynamics model. Intuitively, such a representation contains information only relevant to environment features that the agent can control and features that can have an effect on the agent. RIDE does not consider goal-conditioning, so we do not include language instructions in the observation provided to RIDE.

RIDE's intrinsic rewards are equal to the L2 norm of the difference in this state representation between time steps, scaled by a coefficient $\lambda_R$. We find that $\lambda_R$ can have a sizeable effect on the performance of RIDE, so we tune this hyperparameter as discussed in Section B.3.

We experiment with RIDE in several of the BabyAI tasks, and examine the SEQUENCE-MAZE task as a case-study below. In Figure 3, we compare RIDE with various values of $\lambda$ to ELLA. The SEQUENCE-MAZE task is extremely sparse, and the language instructions are highly diverse. Both RIDE, which rewards impactful state changes, and ELLA, which rewards completion of low-level tasks, have a positive effect on sample efficiency. We additionally test how ELLA+RIDE can be combined; to do this, we simply sum the shaped rewards via ELLA and RIDE at each time step. For this task, we see that the combination of subtask-based exploration (based on an online-learned model of language abstractions) and impact-based exploration (based on online-learned dynamics models) leads to a further increase in sample efficiency.

While these two methods reward different aspects of exploration, and combining them has the potential to improve upon the individual methods, a limitation of this approach is that we must resort to ad hoc methods for tuning shaped reward weights in the combined version. The discussion in Section 4.3 on selecting ELLA's $\lambda$ hyperparameter does not apply to reward functions that are non-sparse, which occurs when additional intrinsic rewards are included.

In SEQUENCE-MAZE, tuning $\lambda$ and $\lambda_R$ for ELLA and RIDE in isolation and then adding the shaped rewards was effective, but this does not hold for all of the environments. As a representative example of this case, Figure 3 compares a tuned version of ELLA+RIDE in with $\lambda = 0.1$ and $\lambda_R = 0.05$ for the Unlock-Maze task. Here, summing the methods did not significantly outperform either individual method. We observe similar behavior in the PUTNEXT-ROOM environments. Future work could examine how best to fuse subtask-based exploration with intrinsic motivation, or how to weigh different types of intrinsic rewards.

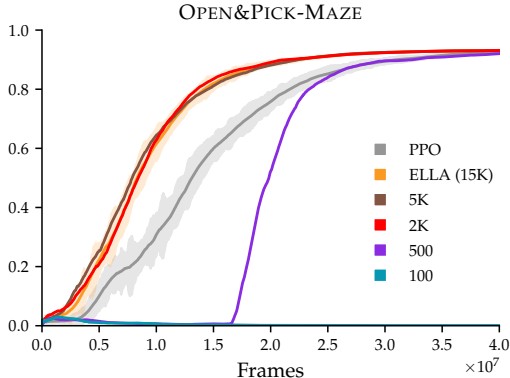

Figure 4: Learning curves in the OPEN&PICK-MAZE environment for vanilla PPO and ELLA with different data budgets for the termination classifier.

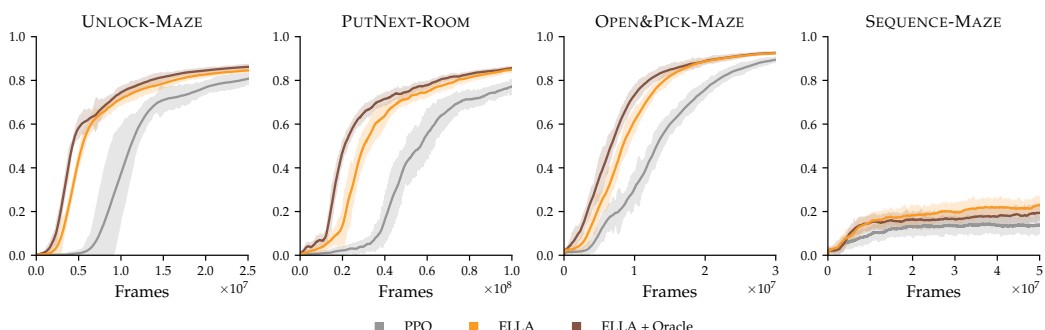

Figure 5: An ablation of ELLA's relevance classifier, where it is replaced with an oracle, in comparison to ELLA and PPO (no shaping).

# E Ablations

Our approach has two learned components: the termination classifier $h^\phi$, which we train offline using annotated examples of low-level termination states, and the relevance classifier $f^\rho$, which is trained online during RL. To understand how these two modules impact performance, we perform ablations on the termination classifier (by varying its offline data budget) and the relevance classifier (by replacing the online version with an oracle).

## E.1 Termination Classifier

One of the assumptions of our method is that examples of low-level termination states are inexpensive to collect. As noted in Section B.1, we train the termination classifier in ELLA with 15K positive and negative examples, such that it achieves 99% accuracy on a balanced validation set. We perform an ablation on the termination classifier by training the termination classifier with lower data budgets, and find that we can reduce the data budget to 2K without affecting ELLA's performance. However, with a data budget as low as 500, ELLA's performance declines. With a data budget of 100, the termination classifier has an accuracy of only 66%, and the shaped rewards are noisy enough that the overall learning curve is degenerate.

## E.2 Relevance Classifier

The relevance classifier is trained online using the relevance dataset $\mathcal{D}$. As an ablation for this module, we replace the relevance classifier with an oracle. To mimic the way that the relevance classifier is initialized to predict that all low-level tasks are relevant to a high-level instruction, we reward all low-level tasks for the first 2.5 million frames before switching to the oracle predictions. This initial phase incentivizes low-level task completion generally, as ELLA does, and is empirically beneficial.

Figure 5 shows learning curves for PPO (no shaping), ELLA, and ELLA with the oracle relevance classifier. The oracle version slightly outperforms ELLA for three of the four tasks and performs similarly on the SEQUENCE-MAZE task. This task has over $10^6$ instructions and remains challenging even with low-level bonuses.

## F  Proof of Policy Invariance

In this section, we sketch the proof of the policy invariance of our reward transformation. We begin with the goal-conditioned MDP $\mathcal{M} = (\mathcal{S}, \mathcal{A}, T, R, \mathcal{G}, \gamma)$, where $T : \mathcal{S} \times \mathcal{A} \times \mathcal{S} \to [0, 1]$ and $R : \mathcal{S} \times \mathcal{A} \times \mathcal{S} \times \mathcal{G} \to [0, R_{max}]$. $R$ is sparse. Let $\tilde{\mathcal{M}}$ be an augmented MDP $(\tilde{\mathcal{S}}, \mathcal{A}, \tilde{T}, \tilde{R}, \mathcal{G}, \gamma)$ which stores state histories: that is, $\tilde{s}_t = (s_t, h_{0:t-1})$. $\tilde{T} : \tilde{\mathcal{S}} \times A \times \tilde{\mathcal{S}} \to [0, 1]$ is defined as $T(\tilde{s}_t, a_t, \tilde{s}_{t+1}) = T(s_t, a, s_{t+1}) \cdot 1[h_{0:t} = (h_{0:t-1}, s_t)]$. $\tilde{R}$ is defined similarly to reflect consistency between histories. The transformation from $\mathcal{M}$ to $\tilde{\mathcal{M}}$ does not affect optimal policies because we are simply appending history information to the state without affecting the dynamics. We now use $\tilde{\mathcal{M}}$ (instead of $\mathcal{M}$) to show policy invariance with a given shaped MDP $\mathcal{M}'$, as we describe below.

Consider a shaped MDP $\mathcal{M}' = (\tilde{\mathcal{S}}, \mathcal{A}, \tilde{T}, R', \mathcal{G}, \gamma)$ where $R' : \tilde{\mathcal{S}} \times \mathcal{A} \times \tilde{\mathcal{S}} \to [0, 1]$ is defined as $R'(\tilde{s}_t, a_t, \tilde{s}_{t+1}) = \tilde{R}(\tilde{s}_t, a_t, \tilde{s}_{t+1}) + \ell(\tilde{s}_t, \tilde{s}_{t+1})$ where $\ell : \tilde{\mathcal{S}} \times \tilde{\mathcal{S}} \to \mathbb{R}$ represents the low-level task bonuses (or neutralization penalty) going from state $\tilde{s}_t$ to $\tilde{s}_{t+1}$ as defined in Section 4.3. We first aim to show that the transformation from $\tilde{\mathcal{M}}$ to $\mathcal{M}'$ does not introduce new optimal policies—that is, any optimal policy in $\mathcal{M}'$ is also optimal in $\tilde{\mathcal{M}}$.

Let $\hat{\pi}_{\tilde{\mathcal{M}}}(\tilde{s}) = \pi^*_{\mathcal{M}'}(\tilde{s})$ where $\pi^*_{\mathcal{M}'}$ is optimal in $\mathcal{M}'$. We will show this policy is also optimal in $\tilde{\mathcal{M}}$: that is, $V^{\hat{\pi}_{\tilde{\mathcal{M}}}}_{\tilde{\mathcal{M}}}(\tilde{s}_t) = V^*_{\tilde{\mathcal{M}}}(\tilde{s}_t)$ for all $\tilde{s}_t$. Since $\tilde{R}$ is nonnegative and sparse, we only need to consider states $\tilde{s}_t$ at which the value could possibly be positive: those from which the task is solvable in at most $H - t$ steps, where $H$ is the horizon.

Assume the task is solvable in a minimum of $k \leq H - t$ steps (using an optimal policy in $\tilde{M}$). We can reason about $\hat{\pi}_{\tilde{\mathcal{M}}}(\tilde{s}) = \pi^*_{\mathcal{M}'}(\tilde{s})$ by considering the various ways return could be accumulated in $\mathcal{M}'$, and which of those cases yields the maximum return.

(1) A policy could solve the task in $j \geq k$ steps while solving subtasks at timesteps $T_{\mathbb{S}}$, and receive a discounted future return of $\sum_{t' \in T_{\mathbb{S}}; t' \geq t} \gamma^{t'-t} \lambda + \gamma^j (R_{max} - \sum_{t' \in T_{\mathbb{S}}} \gamma^{t'-(t+j)} \lambda)$.

(2) A policy could solve only subtasks at timesteps $T_{\mathbb{S}}$ and receive a discounted future return of $\sum_{t' \in T_{\mathbb{S}}; t' \geq t} \gamma^{t'-t} \lambda$.

We can simplify the return in case (1) to $\gamma^j R_{max} - \sum_{t' \in T_{\mathbb{S}}; t' < t} \gamma^{t'-t} \lambda$. The second term does not depend on actions taken after $t$; thus, this case is maximized by completing the task in $j = k$ steps.

Note that case (2) always gets smaller return than case (1): the first term, $\sum_{t' \in T_{\mathbb{S}}; t' \geq t} \gamma^{t'-t} \lambda$, is the same as in case (1), and the second term in case (1) is strictly positive when we use the bound on $\lambda$ from Section 4.3: that $\lambda < \frac{\gamma^H R_{max}}{|\mathcal{G}_\ell|}$.

Therefore, the maximum future return is achieved in case (1), specifically by a policy that solves the task in $k$ steps, and we know that this policy exists. Thus $V^{\hat{\pi}_{\tilde{\mathcal{M}}}}_{\tilde{\mathcal{M}}}(\tilde{s}_t) = \gamma^k R_{max} = V^*_{\tilde{\mathcal{M}}}(\tilde{s}_t)$, and so an optimal policy in $\mathcal{M}'$ is also optimal in $\tilde{\mathcal{M}}$. In order to show the reverse, we can use similar logic to show that any optimal policy in $\tilde{\mathcal{M}}$ acts optimally in $\mathcal{M}'$ for any state $\tilde{s}_t$ where the task is solvable in a minimum of $k \leq H - t$ steps; actions towards solving the task most quickly are also optimal in $\mathcal{M}'$. Together, this shows policy invariance between $\tilde{\mathcal{M}}$ and $\mathcal{M}'$ for the set of states $\tilde{s}_t$ where the task is solvable.