# OpenReview forum: "ELLA: Exploration through Learned Language Abstraction"
_NeurIPS.cc/2021/Conference — NeurIPS 2021 Poster_

### Official Review · Reviewer_3GCC · 2021-07-13

**Rating:** 7
**Confidence:** 4

**Summary:**

The paper present ELLA, a three-part solution to instruction following tasks with a high-level instruction and a pre-defined set of low-level instructions:

1. a termination classifier is trained offline to identify the completion of a low-level goal at a state;
2. a relevance classifier is learned online to predict a binary relevance of a low-level goal to a high-level goal.
3. reward shaping is used to encourage low-level goals suggested by the relevance classifier.

ELLA is shown to improve sample complexity on 5/6 BabyAI tasks of diverse environment and instruction complexities, compared to an ablation PPO model with reward shaping, and a baseline model LEARN.

**Limitations And Societal Impact:**

Cons/questions:

- The baselines might be under-explained and a bit weak. According to my understanding, the PPO baseline makes use of neither high-level or low-level instructions, and the LEARN baseline just uses the high-level instruction? More detailed setups are needed. And it might be a less surprising comparison if ELLA is the only model that leverages the low-level instruction set. See below for some suggested baselines/model variants.

- A major model design is to learn termination classifier offline and relevance classifier online. Is it possible to show the opposite, e.g. learn the termination classifier online, while learning the relevance classifier offline (possible since BabyAI is synthetic)?

- Another potential concern is, learning relevance classifier totally offline abandons the fact that instructions are languages with compositional structures. For example, a high-level goal “goto box and pick it up” can be easily decomposed to “goto box” and “pickup box”, even without the game context. Currently, for each game, a new relevance classifier is learned from scratch about such decompositions, and no language priors is used. I wonder if this classifier can be somehow pre-trained as well and boost performance.

- Another model design is reward shaping. Is it possible not to use it, but directly learn a relevance classifier plus a low-level goal-conditioned policy? Some experiments are needed to justify reward shaping compared to such an approach without it.

- I don’t totally get Section 6.2. “However, the large number of diverse language instructions makes it difficult to estimate good decompositions via our deduplication method, and so it is difficult to learn f ρ online and offer the most targeted low-level task bonuses.” So for the PutNext task, why sample efficiency is still improved even if the relevance classifier does not learns good decompositions?

- The relevance classifier seems to have no explicit temporal order of decompositions. Is this a concern for more practical tasks, or is it solid because only one relevant low-level goal can be achieved any time? Some discussion is needed.


**After rebuttal**

Thanks to authors for clarifying my questions. The baselines are stronger than I thought, and it is reasonable to adopt some of my advice as future work. I raise the score to 7 and hope the paper can add/clarify discussion with respect to my questions.

**Main Review:**

Pros:

- The method is well-motivated for the considered task setup of three levels: high-level instruction, low-level instructions, and low-level actions. Such a setup is natural for many real-world language-guided tasks, since language is hierarchical. Directly mapping the compositional high-level instruction to low-level actions is sample-inefficient, and this is a fundamental problem in instruction following. To make use of a pre-defined set of low-level instructions, a method must
A. Link low-level instructions to low-level actions. This is done by the termination classifier providing pseudo-rewards, and reward shaping to affect the low-level action policy.
B. Link high-level instruction to low-level instructions. This is done by the relevance classifier learned online.

- It is infeasible to learn both A and B online, so ELLA proposes to learn the termination classifier offline and the relevance classifier online, with the assumption that the former supervision is easier to collect. The consequent empirical results look reasonable.

- The paper is well written with good figures.


**Time Spent Reviewing:**

4

---

> ### Author Response · Authors · 2021-08-10
> **Response to Reviewer 3GCC**
>
> Thank you for your detailed review. We appreciate the comments that our work is “well-motivated,” addresses a “fundamental problem,” and is “well-written.” We respond to your comments and questions below.
>
> **1) Baselines**
>
> We will move some of the exposition of the baselines from the Appendix to the main paper.
>
> To clarify the setup, the PPO baseline _does_ in fact make use of the high-level instructions (by encoding both the game state and the language instruction), just as LEARN, ELLA, RIDE, and ELLA+RIDE do. LEARN additionally does makes use of low-level instructions, in that its relevance classifier is pre-trained using low-level instructions and corresponding demonstrations. Our goal was to include baselines that leverage both high-level and low-level instructions, and we believe our current set of baselines achieves this goal.
>
> **2) Online & Offline Learning**
>
> It may be possible to learn a relevance classifier offline. However, learning the termination classifier online would be tricky. In the strict hierarchical case, one could learn distinct options/subpolicies (Andreas et al. 2017) given the recipe of relevant low-level instructions. However, since we do not make the assumption of a strict hierarchy, and only that high-level tasks are supported by low-level tasks (L131-138), it is possible that the low-level tasks overlap in time or require primitive actions between them. Learning the termination classifier online in this case is an interesting question for future work.
>
> **3) Language Priors**
>
> Indeed, the example “goto box and pick it up” can be easily decomposed to “goto box” and “pickup box.” However, ELLA is capable of performing decompositions that are not as explicit, and require some amount of grounding in the game context (e.g. “open the yellow door” has “pick up the yellow key” as a low-level task). We believe that an online learning component for the relevance classifier online is important to discovering and learning such decompositions.
>
> Additionally, we do agree that instilling language priors via offline pre-training could further boost the performance of our method; we are happy to add additional discussion on this point.
>
> **4) Reward Shaping**
>
> We cannot directly use the suggested setup of a relevance classifier plus a low-level goal-conditioned policy because of the general way in which we define the relationship between high- and low-level tasks: specifically, high-level tasks need not consist of an exact sequence of low-level tasks.
>
> **5) Section 6.2 Clarification**
>
> We will adjust the wording to clarify this point. In the sentence you quote from L329-331, we are referring specifically to the Sequence-Maze task. There is some gain in sample efficiency (which can be attributed to the idea that incentivizing completion of meaningful behavior — both relevant and irrelevant to the high-level task — can lead to some exploration boost). However, since Sequence-Maze has over 1 million diverse high-level instructions, seeing the same instruction more than once is very rare, and so it is difficult to deduplicate the labels for the online relevance dataset. Learning good decompositions is difficult in this case.
>
> In the PutNext-Maze task, as seen in Figure 5, the relevance classifier does learn good decompositions; it reaches nearly 100% accuracy. There is a strong increase in sample efficiency in this case (Figure 3).
>
> **6) Temporal Order**
>
> The relevance classifier does not model the temporal order of the low-level tasks. This is not explicitly limiting: providing intermediate bonuses for low-level tasks that an agent completes out of order may be helpful for exploration. Modeling the order of relevant abstractions would be an interesting component of future work.
>
> To the second question: ELLA can in fact handle the case that multiple relevant low-level tasks terminate at a given time step. For example, “go to the red ball” and “go to a red ball” both terminate at the same time when performing the high-level task “pick up the red ball.” We will include a note about this capability in the text.

---

> > ### Author Response · Authors · 2021-08-26
> > **Follow-up to Reviewer 3GCC**
> >
> > Thank you again for your review. We wanted to quickly follow up and see if our response adequately addresses your questions and comments, or if you have additional concerns.

---

### Official Review · Reviewer_Yaq9 · 2021-07-16

**Rating:** 6
**Confidence:** 4

**Summary:**

This paper presents a reward shaping approach that provides immediate rewards through the achievement of low-level sub-tasks. The authors assume a problem where a high-level task given in the natural language in a sparse reward environment consists of several low-level sub-tasks, and aim to guide the agent to effectively explore using task structure. Two components (termination classifier and relevance classifier) are proposed, and reward shaping is performed using them. Finally, the authors demonstrated the results on BabyAI, a grid world platform for instruction following tasks.

**Limitations And Societal Impact:**

Yes, the authors have adequately addressed it.

**Main Review:**

I agree that this is an interesting and important problem. However, I think the assumptions in the paper are quite strong, and I am concerned about whether the experiments shown in the paper demonstrate the effect of language abstraction well. Comparison with additional baselines should be included, and it seems that the experiment should be modified to better reveal the effect of language abstraction. The detailed comments and questions are as follows:

1. The baseline algorithms seem to be poor with PPO and LEARN. ELLA aims to effectively perform the exploration using language abstraction, then I think it should show advantages and improvements compared to existing exploration methods that do not use the information of language abstraction (ex. uncertainty-based exploration [1], curiosity-based exploration [2], or impact-driven exploration [3]). Some tasks were compared with RIDE in the appendix, but I think that the results on other tasks should also be mainly compared and addressed in the main text.

2. ELLA uses more information than RIDE (instruction set of subtask and human-labeling of termination states), but despite this, RIDE and ELLA show similar results in terms of sample efficiency and performance. The current experimental results do not reveal the advantages of ELLA compared to the existing approaches that do not use language abstraction information. It is thought that the generalization effect of language abstraction can be shown more meaningfully by conducting experiments on unseen tasks such as meta-learning or transfer learning settings.

3. As a weakness of this paper, I am concerned with the following two points: 1) The assumption for termination states of instructions are quite strong. In the general case, it is very expensive to label a large number of data manually. 2) It seems that performance and sample efficiency are sensitive to $\lambda$ parameters.

4. (Page 9, lines 310-313) I don't understand how the process of calculating the $\lambda$ is done. How is $\lambda$ computed from step here?

5. (Page 8 lines 281-285) The authors explain why ELLA does not increase sample efficiency in a COMBO environment, but I don't quite understand what it means.

[1] Yuri Burda et al, Exploration by Random Network Distillation, ICLR 2019

[2] Deepak Pathak et al, Curiosity-driven Exploration by Self-supervised Prediction, ICML 2017

[3] Roberta Raileanu et al, RIDE: Rewarding Impact-Driven Exploration for Procedurally-Generated Environments, ICLR 2020


**Time Spent Reviewing:**

10

---

> ### Author Response · Authors · 2021-08-10
> **Response to Reviewer Yaq9**
>
> Thank you for your detailed review. We respond to each of your comments and questions below.
>
> **1 & 2) Existing Exploration Methods**
>
> Thank you for the suggestion! In examining the relationship of ELLA to intrinsic motivation methods in Appendix D, we specifically chose impact-driven exploration (RIDE) due to its superior performance on MiniGrid tasks compared to curiosity-based exploration (ICM) and uncertainty-based exploration (RND).
>
> RIDE and ELLA indeed reward different types of exploration — the former based on learned dynamics models rather than language abstractions — which is why we believe such methods are orthogonal to ELLA. Depending on the task, language-subgoal-based exploration may become more beneficial when combined with intrinsic motivation methods. In this vein, we conducted a few experiments on stacking the two types of exploration. We are happy to expand these experiments and include the results in the main text (space permitting) or the Appendix.
>
> **3.1) Assumption of Termination States**
>
> Building upon prior work, we assume these examples are not too difficult to obtain (Bahdanau et al, 2019); low-level tasks are simpler and less compositional. Prior work such as Jiang et al. (2019) make an even stronger assumption that an oracle termination function is given; Waytowich et al. (2019), Goyal et al. (2019), and Lynch et al. (2021) make a similar assumption to us that demonstrations are available. Thus we believe this is not a strong assumption for our setting.
>
> To understand the effect of different data budgets for termination classifier in our domain, we perform an ablation on the data budget in Appendix E.1. In domains where collecting low-level demonstrations is costly, we agree that our approach is less feasible (see Section 7).
>
> **3.2) Sensitive to** $\lambda$
>
> We compare a wide range of values for $\lambda$ as evidence for the idea that egregious violations of policy invariance lead to unstable learning. While further tuning may still be necessary, Section 4.3 provides guidelines for choosing a reasonable range for $\lambda$.
>
> **4) Computation of $\lambda$**
>
> For PutNext-Room with GoTo-Room as the low-level task, we have that $H = 128$ and $|\mathcal{G}_\ell| = 36$. (The latter value is in Appendix A, and we will add the former value to the paper). In BabyAI, the reward for finishing a task at time step $t$ is $20 (1-0.9 * \frac{t}{H})$ (see L251-252). So $r_H$, the reward for finishing the task at time step $H$, is $20 (1-0.9(128/128)) = 2$.
>
> Then, plugging into inequality (1), we have that $\lambda < \frac{\gamma^H r^H}{|\mathcal{G}_\ell|} = \frac{0.99^{128}*2}{36} = 0.0153$. This justifies our selection of $\lambda= 0.015$. Under the assumption that the expert policy solves the task in under 100 steps (we have a tighter bound for $M$ than $H$) we can perform a similar calculation to the above and to get $\lambda < 0.06$. This justifies our selection of $\lambda = 0.05$. We will add clarity to Section 6.1.
>
> **5) COMBO Environment**
>
> The COMBO environment includes instructions such as “put the red ball next to the green box,” “open the green door,” and “pick up the blue box.” For the latter two instruction types (open X and pick up X), there is only one additional action that an agent must make after navigating to X in order to solve the task (namely, the “open” or “pick up” action). In other words, the low-level task GoTo is similar to the high-level tasks. Therefore, the improvement in exploration by receiving an intermediate reward from “going to” the relevant object is relatively small compared to just receiving the final reward for completing the high level task. We are happy to clarify these lines in the paper.

---

> > ### Comment · Reviewer_Yaq9 · 2021-08-26
> > **Response to rebuttal**
> >
> > Thank you for providing the rebuttal to respond to my questions and concerns. Most of my questions and concerns have been addressed, but I am still concerned with strong assumptions of termination state and access to a set of low-level language instructions. As for the other parts, it seems to be no problem if the paper is re-organized based on the rebuttal.
> >
> > Except for strong assumptions, I think that the proposed method is well-motivated and reasonable, so I am willing to increase my score to 6. I believe that re-writing and re-organizing the paper based on what was mentioned in the rebuttal can make it more convincing.

---

> > > ### Author Response · Authors · 2021-08-26
> > > **Follow-up to Reviewer Yaq9**
> > >
> > > We are glad to hear that our response addressed most of your questions and concerns. We will revise the paper based on your suggestions. Thanks again.

---

### Official Review · Reviewer_HPzX · 2021-07-18

**Rating:** 5
**Confidence:** 2

**Summary:**

This paper suggests ELLA (Exploration through Learned Language Abstraction), a reward shaping approach for language instruction folllowing tasks. The contributions that the authors argue in this paper can be summarized as follows:
1. propose a reward shaping approach for instruction following, which consist of two main module, (1) termination classifier and (2) relevance  classifier
2. empirically show that the proposed method is effective by experiment on BabyAI environment, which is a representative benchmark for instruction following tasks.

**Limitations And Societal Impact:**

Yes, the authors adequately have addressed the limitations and potential negative societal impact of their work.

**Main Review:**

This paper presents an exploration strategy of reinforcement learning agents to solve instruction following tasks. Since instruction following tasks are usually specified on a sparse reward setting, reward shaping can be effective for reinforcement learning. I think the most strong point of ELLA is a policy invariance property, which is usually not guaranteed on recent intrinsic exploration approaches.

However, I have following concerns and questions for this papers:

1. Although this paper consider on an exploration strategy for language guided tasks, I think considerations of this approach seems task-specific rather than language-specific. The authors especially only consider hierarchical tasks where a high-level instruction (goal) is able to be decomposed into low-level instructions (sub-goals).

2. In Table 1 in Appendix A, what are criteria of assigning low-level tasks for each high-level task?

3. Can you clarify how is low-level instructions data (used in experiments) collected? Whether low-level instruction data is collect by human  demonstration or demonstrations generated by human-crafted oracle, I understand each low-level instruction dataset should be collected individually for each low-level task. I think this requirement is also too expensive for general applications.

4. Figure 4 shows ELLA is sensitive with a hyperparameter selection in that inappropriate selected $\lambda$ can make learning failure. This also can be one of limitations of this paper.


**Time Spent Reviewing:**

12

---

> ### Author Response · Authors · 2021-08-10
> **Response to Reviewer HPzX**
>
> Thank you for your comments and questions. We provide a response to each of your points below.
>
> **1) Problem Setting**
>
> Our main assumption is that a high-level instruction can be at least partially decomposed into simpler steps. This is not a limiting assumption—indeed, as one starts to study more complex tasks beyond achieving simple flat objectives, compositionality will become more prevalent. Consider a high-level instruction such as “unlock the door”; there are low-level tasks such as “find the key” which are implicit in the instruction, and so our assumption is satisfied. This does not make our approach task-specific beyond the fairly general assumption of compositionality. We would also stress that our notion of language abstraction is _flexible_ compared to prior work (Jiang et al. 2019, Andreas et al. 2017), in that task decompositions need not be known _a priori_, and crucially, that decompositions need not be strict. We agree with Reviewer 3GCC that our setup “is natural for many real-world language-guided tasks.”
>
> **2) Criteria for Low-Level Tasks**
>
> We select all of our task families from the BabyAI platform, using either pre-defined or custom tasks. For each high-level task family, we hand-select one or more low-level task families which are relevant to the high-level tasks. An additional consideration in choosing the pairings was to vary along the axes discussed in Section 5: sparsity of the high-level task, similarity of the low- and high-level tasks, and the number of low-level tasks relevant to each high-level task.
>
> **3) Low-Level Data**
>
> We use the automated expert in BabyAI to generate low-level trajectories, as we note in Appendix B.1. We assume such examples are easy to obtain (Bahdanau et al, 2019); low-level tasks are simpler and less compositional. This assumption is fairly common in prior work. For example, Jiang et al. (2019) make a stronger assumption that an oracle termination function is given; Waytowich et al. (2019), Goyal et al. (2019), and Lynch et al. (2021) make a similar assumption to us that demonstrations are available, with Lynch et. al. showing that collecting language for such "simple demonstrations" is cheap and scalable.
>
> To understand the effect of different data budgets for termination classifier in our domain, we perform an ablation on the data budget in Appendix E.1. In domains where collecting low-level demonstrations is costly, we agree that our approach is less feasible (see Section 7).
>
> **4) Sensitive to** $\lambda$
>
> As we note in Section 6.1, values of $\lambda$ that egregiously violate policy invariance can lead to learning failure. We emphasize that we specifically chose to experiment with such values in order to demonstrate this claim. We believe that these results provide a strong rationale for using the guidelines for choosing $\lambda$ as specified in Section 4.3. These guidelines can help lead to reasonable values, as we discuss in Section 6.1. We are excited to be able to analyze the effects of this hyperparameter rather than leaving it to the user to choose.

---

> > ### Author Response · Authors · 2021-08-26
> > **Follow-up to Reviewer HPzX**
> >
> > Thank you again for your review. We wanted to quickly follow up and see if our response adequately addresses your questions and comments, or if you have additional concerns.

---

> > > ### Author Response · Authors · 2021-08-31
> > > **Re: Follow-up to Reviewer HPzX**
> > >
> > > We understand that this review format is taxing, so we've strived to crisply address all the points in your review. We hope that if there are any remaining questions or additional feedback you have, we can work together to answer them, and make this paper the best it can be!

---

### Author Response · Authors · 2021-08-10
**Summary Response**

Thank you all for the comments. We appreciate the feedback that our work is “well-motivated,” addresses a “fundamental problem” that is "interesting and important," has "reasonable empirical results," and is “well-written with good figures.”

The core idea of ELLA is to leverage the abstractive property of language to guide exploration in RL for instruction following without the strict assumptions of prior work. Given the severe sample inefficiency of language+RL methods, we strongly believe that our work is a step in the right direction. ELLA rewards a unique aspect of exploration that would be a useful companion to current methods.

We have addressed each of the questions and concerns point-by-point in separate responses below, and would be happy to discuss further as needed.

---

### Decision · Program_Chairs · 2021-09-27

**Decision:**

Accept (Poster)

**Comment:**

The paper describes an approach to training an agent to follow instructions for tasks composed of multiple low-level subtasks. The method uses reward shaping, whereby the agent receives supplementary rewards when sub-tasks are achieved. The shaped rewards are determined using two classifiers, a "terminal classifier" that classifies the completion of a low-level subtask, and a "relevance classifier" that assesses the relationship between low-level tasks and the success of the high-level task. The latter is learned online without the need to break high-level instructions into their lower-level components. Experimental results on the the BabyAI task and a grid world task demonstrate gains in sample efficiency relative to recent baselines.

The reviewers agree that the paper considers an important and challenging domain. The problem of understanding natural language instructions has long been the focus of AI and robotics research, and has recently received renewed attention within the broader machine learning community. The idea of exploiting the hierarchical nature of the tasks as a means of encouraging exploration through reward shaping is reasonable and well motivated. The reviewers point out that the approach relies on several strong assumptions (e.g., involving the existence of terminal states and access to low-level instructions). Two of the reviewers also raise concerns about the baselines and the amount of information that they are provided with compared to ELLA. The author response helped to resolve some of these concerns, but the authors are encouraged to ensure that they are addressed in any subsequent version of the paper.